# The C2H2 Zinc Finger Protein MaNCP1 Contributes to Conidiation through Governing the Nitrate Assimilation Pathway in the Entomopathogenic Fungus *Metarhizium acridum*

**DOI:** 10.3390/jof8090942

**Published:** 2022-09-07

**Authors:** Chaochuang Li, Yuxian Xia, Kai Jin

**Affiliations:** 1Genetic Engineering Research Center, School of Life Sciences, Chongqing University, Chongqing 401331, China; 2Chongqing Engineering Research Center for Fungal Insecticide, Chongqing 401331, China; 3Key Laboratory of Gene Function and Regulation Technologies Under Chongqing Municipal Education Commission, Chongqing 401331, China

**Keywords:** *Metarhizium acridum*, C2H2 zinc finger protein MaNCP1, nitrate assimilation, conidiation, *MaAreA*

## Abstract

Zinc finger proteins are an important class of multifunctional regulators. Here, the roles of a C2H2 zinc finger protein MaNCP1 (*Metarhizium acridum* nitrate-related conidiation pattern shift regulatory factor 1) in nitrogen utilization and conidiation were explored in the entomopathogenic fungus *M**. acridum*. The results showed that *MaNCP1*-disruption mutant (Δ*MaNCP1*) impaired the ability to utilize nitrate, ammonium and glutamine and reduced the expression of nitrate assimilation-related genes, suggesting that MaNCP1 was involved in governing nitrogen utilization. In addition, the conidial yield of the Δ*MaNCP1* strain, cultured on the microcycle conidiation medium (SYA), was significantly decreased, which could be restored or even enhanced than that of the WT strain through increasing the nitrate content in SYA medium. Further study showed that *MaAreA*, a core regulator in the nitrogen catabolism repression (NCR) pathway, was a downstream target gene of MaNCP1. Screening the differential expression genes between WT and Δ*MaNCP1* strains revealed that the conidial yield of *M. acridum* regulated by nitrate might be related to NCR pathway on SYA medium. It could be concluded that *MaNCP1* contributes to the nitrate assimilation and conidiation, which will provide further insights into the relationship between the nitrogen utilization and conidiation in fungi.

## 1. Introduction

Crop pests seriously threaten agricultural production, and are important factors restricting global food production and quality, causing huge losses to agriculture every year [1,2]. Therefore, the effective control of crop pests is the important link in food security production [3,4]. Entomopathogenic fungi are the natural pathogenic microorganisms of insects and exhibit great potentials in agricultural pest control. For example, *Beauveria bassiana* and *Metarhizium anisopliae* have been developed into pesticides and applied in pest control [5]. For fungal pesticides, conidia are the main effective components to adhere to host cuticles, and germinate under certain conditions to form appressoria, then forming infection pegs to penetrate host cuticles and enter into hemolymph, and finally destroying immune systems and killing host pests [6]. Conidial production is one of the main restrictive factors for the large-scale application of fungal pesticides [7,8]. Thus, elucidating the regulatory mechanism of conidiation in entomopathogenic fungi will be helpful to fully tap their potentials in agricultural pest control.

Asexual sporulation is the main reproductive mode of filamentous fungi. At present, numbers of sporulation-related genes have been characterized, *BrlA*, *AbaA*, and *WetA* are the central regulatory genes in the conidiation regulatory pathway in *Aspergillus nidulans* [9,10,11], mutations in any of these three regulatory factors will block the development of conidia [9]. *fluG*, *flbA*, *flbB*, *flbC*, *flbD* and *flbE* play positive roles in regulating the central regulatory pathway, which were known as the upstream developmental activators (UDAs), and are mainly to activate *brlA* and initiate conidial formation [12,13]. FluG, locates at the most upstream of UDAs, is a key activator for the initiation of conidia in *A. nidulans* [14]. In fact, asexual sporulation can be affected by many factors, among which nutritional condition, such as nitrogen source [15,16,17]. However, the relationship between the nitrogen utilization and conidiation in fungi is not fully understood in fungi.

For most filamentous fungi, they also have a further conidiation pattern, that is, microcycle conidiation [18]. Previous results have shown that the microcycle conidiation medium (SYA) can induce insect pathogenic fungus *M. acridum*, an acridid-specific pathogen, to perform microcycle conidiation [19], which would promote the fungal conidiation and increase the conidial yield, and these conidia exhibited higher thermotolerance and similar UV-B tolerance and virulence [20]. Meanwhile, increasing the nitrate in SYA medium could shift the conidiation pattern in *M. acridum* [21], suggesting that nitrate metabolism may play important roles during this process.

In fungi, nitrate assimilation plays important roles in nitrate utilization and is regulated by the NCR path, which is mediated by GATA transcription factors [22]. AreA and AreB, the only two GATA factors in NCR path are characterized in filamentous *Ascomycetes*, and the core transcriptional activator AreA could work synergistically with NirA to promote nitrate assimilation in *A. nidulans* [23]. Under nitrogen starvation conditions, higher *areA* transcription levels [24], more stable mRNA levels [25,26] and preferential nuclear localization of protein [27] would increase the abundance and activity of AreA. Furthermore, AreA could interact with the negative regulatory factor NmrA to decrease the abundance and activity of AreA when the nitrogen is sufficient [28,29].

Recently, MaNCP1 contributes to the conidiation pattern via nitrate assimilation was identified [30] and its N-terminal zinc fingers make contributions to the fungal growth and virulence in *M. acridum* [31]. However, the roles of MaNCP1 in conidiation need to be further explored. Here, we focused on the relationship between nitrate assimilation and conidiation. These data revealed that *MaNCP1* contributes to the conidiation through governing the nitrate assimilation pathway, which will broaden the research field of nutritional regulation conidiation in filamentous fungi.

## 2. Materials and Methods

### 2.1. Strains and Culture Conditions

All *MaNCP1*-relatived mutants have been generated in our previous study [30], and cultured on 1/4SDAY (2.5‰ peptone, 5‰ yeast extract, 10‰ glucose and 18‰ agar, *w*/*v*), SYA (30‰ sucrose, 5‰ yeast extract, 0.5‰ MgSO_4_, 1‰ KH_2_PO_4_, 0.5‰ KCl, 0.01‰ MnSO_4_, 0.01‰ FeSO_4_, 3‰ NaNO_3_ and 18‰ agar, *w*/*v*), SYA+N (SYA supplemented with 40.5‰ NaNO_3_, *w*/*v*), Czapek-dox (CZA) (0.01‰ FeSO4, 0.5‰ KCl, 0.5‰ MgSO_4_, 1‰ K_2_HPO_4_, 2‰ NaNO_3_, 30‰ sucrose and 18‰ agar, *w*/*v*) and/or modified CZA medium at 28 °C for days. The *M. acridum* strain CQMa102 (wild-type, WT) was deposited in China General Microbiological Culture Collection Center (CGMCC, Beijing, China; No. 0877). *Escherichia coli* DE3 (Solarbio, Beijing, China) was used for the prokaryotic expression of the target proteins. Y187 (Clontech, Palo Alto, CA, USA) yeast strain was used for yeast one-hybrid assay.

### 2.2. Conidial Yield and Fungal Growth Assays

Conidial suspensions with a concentration of 10^6^ conidia/mL were prepared after the fungal strains grown on 1/4SDAY for 15 d, pipetting 2 μL and, respectively, inoculated into 24-cell plates contain SYA, SYA + N, CZA or modified CZA media and followed by growing at 28 °C to determinate the conidial yield [32]. Two microliters conidial suspensions were, respectively, inoculated into CZA or modified CZA with different nutrients and cultured at 28 °C for 7 d to observe the growth of each fungal strain.

### 2.3. Bioinformatics Analysis and Yeast One-Hybrid Assay

The putative binding *cis*-element of MaNCP1 was analyzed with JASPAR 2020 [33]. The cDNA of MaNCP1 and the promoter sequence of *MaAreA-1* (−2019~−996) and *MaAreA-2* (−1013~+51) were ligated into pGADT7 or pHIS2 vector to generate pGADT7-MaNCP1, pHIS2-*MaAreA-1* and pHIS2-*MaAreA-*2, respectively. 3-amino-1,2,4-triazole (3-AT), a competitive inhibitor, was added into selected medium SD/-Trp/-His (DDO) to inhibit the leakage expression of HIS3. Co-transforming pGADT7-MaNCP1 with pHIS2-*MaAreA*-*1/2* into Y187 to obtain Y187 (pGADT7-MaNCP1×pHIS2-*MaAreA-1*/2), then spread on SD/-Leu/-Trp/-His (TDO) and TDO contained 3-AT with the control group setting: positive controls (pGADT7-53 and pHIS2-53) and negative control (pGADT7-MaNCP1 and pHIS2).

### 2.4. Electrophoretic Mobility Shift Assay (EMSA)

The expression and purification of the N-terminal of MaNCP1 protein (228 aa) were performed according to our previously study [30]. The DNA probe was cloned with primer pairs AreA-P-F/AreA-P-R (Appendix A). EMSA Probe Biotin Labeling Kit (Beyotime, Shanghai, China) and Chemiluminescent EMSA kit (Beyotime, Shanghai, China) were used for labeling the probe with biotin and EMSA, respectively [30].

### 2.5. Quantitative Reverse Transcription PCR (qRT-PCR) Analyses

Fungal strains, cultured on SYA or SYA+N plates at 28 °C for 12, 18 or 24 h, were harvested and used for extracting RNA with Fungal RNA Kit (OMEGA, USA), followed by synthesizing cDNA via PrimeScript^TM^ RT reagent Kit with gDNA Eraser (TaKaRa, Dalian, China), and qRT-PCR were performed with SYBR^®^ Premix Ex Taq^TM^ kit (TaKaRa, Dalian, China). Data were analyzed with the 2^−ΔΔCt^ method [34] with the internal control gene *gpdh* (*MAC_09584*) and the glyceraldehyde 3-phosphate dehydrogenase gene. Primers were listed in Appendix A.

### 2.6. RNA-Seq Analysis

RNA-seq was performed to uncover the regulatory roles *MaNCP1* involved in nitrate metabolism. Fungal RNAs were extracted after the strains growing on SYA+N media at 28 °C for 18 h, and then submitted to BGISEQ-500 sequencing platform (BGI, Shenzhen, China) for RNA-Seq with three biological replicates. Genes with log_2_(∆*MaNCP1*_N/WT_N) ≥ 1 and FDR (false-discover rate) ≤ 0.001 were defined as the DEGs.

### 2.7. Statistical Analysis

The data (mean ± SE) were analyzed using the SPSS 24.0 via ANOVA (one-way analysis of variance) or T-text method with three biological replicates.

## 3. Results

### 3.1. MaNCP1 Regulates the Conidiation

Analysis of the conidial yield of the fungal strains under SYA or SYA+N conditions found that it was significantly decreased in the absence of *MaNCP1* under SYA condition, which could be restored by increasing exogenous nitrate into the SYA medium (i.e., the SYA + N medium) and finally higher than that of the WT (Figure 1), suggesting that the conidiation was mediated by nitrate.

### 3.2. MaNCP1 Regulates the Nitrogen Utilization

The colony of ∆*MaNCP1* that grown on the nitrate or Glu was slightly smaller than that of the WT or CP (complementary) strains, which was significantly larger when Gln as the sole nitrogen source, but with no difference when grown on the (NH_4_)_2_SO_4_ medium (Figure 2A,B). The conidial yield of the ∆*MaNCP1* strain cultured on nitrate, (NH_4_)_2_SO_4_, Gln or Glu media were all significantly reduced, whether they were cultured for 7 d or 15 d (Figure 2C). Interestingly, the color of the media around the colony of ∆*MaNCP1* strain changed to yellow at the 7 day when Gln was used as the nitrogen source, whether culturing on the plates (Figure 2A) or inoculating into the 24-cell plates, in which the growth of the colony was severely inhibited (Figure 2D). Furthermore, the color of the fermentation broth that inoculated with ∆*MaNCP1* strain cultured for 48 h were also changed to yellow (Figure 2D), and the biomass was significantly reduced (Figure 2E). These results indicated that *MaNCP1* was involved in the regulation of nitrogen utilization.

### 3.3. MaNCP1 Regulates the Expression of Nitrate Metabolism Genes

To further explore the function of *MaNCP1* in nitrate assimilation, nine genes, including nitrate transporter gene *MaNrtB* (MAC_03189), nitrate reductase gene *MaNR* (MAC*_*08624), nitrite reductase gene *MaNiR* (MAC_03493), glutamine synthetase 1 gene *MaGS1* (MAC_01108), glutamine synthetase 2 gene *MaGS2* (MAC_06858) and glutamine synthetase 3 gene *MaGS3* (MAC_04461), glutamate synthase gene *MaGOGAT* (MAC_00032), glutamate dehydrogenase 1 gene *MaGDH1* (MAC_08384) and glutamate dehydrogenase 2 gene *MaGDH2* (MAC_01648), in the nitrate metabolism pathway were searched out from *M. acridum* genome. Under SYA culture conditions (Figure 3A), the expression of *MaNrtB* in ∆*MaNCP1* background was significantly down-regulated, indicating that the transmembrane transport of nitrate from extracellular to intracellular was impaired with disruption of *MaNCP1*. The transcription of *MaNR*, *MaNiR*, *MaGS1* and *MaGS2* were all down-regulated, *MaGDH1*, *MaGDH2* and *MaGOGAT* were all up-regulated at 12 h or 18 h, *MaGS3* was down-regulated at 12 h or 18 h. Under SYA+N culture conditions (Figure 3B), genes, except for *MaNiR* and *MaGS1*, were all up-regulated to varying degrees at different time points in ∆*MaNCP1* background, indicating that *MaNiR* and *MaGS1* were *MaNCP1*-dependent expression under SYA + N condition. It confirmed that *MaNCP1* was involved in regulating nitrate assimilation pathway.

### 3.4. MaAreA Is a Target Gene of MaNCP1

AreA is known to play important roles in nitrate metabolism. Bioinformatics analysis found the *MaAreA* promoter contained the recognition site of MaNCP1 (Figure 4A). In addition, the transcription level of *MaAreA* (*MAC_00939*) in ∆*MaNCP1* background was significantly down-regulated under both SYA and SYA+N conditions (Figure 4B), indicating that *MaAreA* was a *MaNCP1*-dependent expression gene. Furthermore, the Y187 (pHIS2-*MaAreA-1*) and Y187 (pHIS2-*MaAreA-2*) yeast strains did not grow on the DDO plate with 8 mM 3-AT (Appendix A), and only the co-transformed strain Y187 (pGADT7-MaNCP1 × pHIS2-*MaAreA-1*) could grow normally on the TDO + 3-AT (8 mM) plate (Figure 4C). Moreover, EMSA confirmed that the N-terminal of MaNCP1 protein could bind to *MaNmrA* (Figure 4D). These results showed that *MaAreA* was a downstream target gene of MaNCP1.

### 3.5. Transcriptomic Insights into Pleiotropic Effects of MaNCP1

To identify the genes and pathways that regulated by MaNCP1, RNA-seq was performed and a total of 9,699 genes were mapped to the genome, and 72 DEGs (up/down ratio, 43:29) were identified (Figure 5A; Appendix A). To verify the dependability of the RNA-seq data, 22 DEGs were randomly selected to detect their expression by qRT-PCR. As a result, the expression patterns of all these DEGs were similar to those from the RNA-seq data (Appendix A). The Gene Ontology (GO) function annotation showed that these DEGs were mostly enriched to biological process and involved mainly in metabolic process, cellular process, localization, response to stimulus, biological regulation, regulation of biological process and signaling; enriched to cellular component and involved mainly in membrane, membrane part, cell and cell part; enriched to molecular function and involved mainly in catalytic activity, binding and transporter activity (Figure 5B). Kyoto Encyclopedia of Genes and Genomes (KEGG) analysis showed that these DEGs enriched to 13 pathways, such as cellular community-prokaryotes, transcription, translation, amino acid metabolism, carbohydrate metabolism, energy metabolism, global and overview maps, glycan biosynthesis and metabolism, lipid metabolism, metabolism of cofactors and vitamins, metabolism of other amino acids, nucleotide metabolism and xenobiotics biodegradation and metabolism (Figure 5C), suggesting that *MaNCP1* was involved in multiple biological and metabolic regulatory processes.

In addition, among these DEGs, 19 genes (up/down ratio, 11:8) encode hypothetical proteins, the other 53 genes were annotated, in which 4 genes were directly involved in nitrogen metabolism (*MAC_05898*, *MAC_04042*, *MAC_09768* and *MAC_07959*), 6 genes were involved in sugar metabolism (*MAC_07957*, *MAC_09405*, *MAC_02819*, *MAC_00175*, *MAC_01184* and *MAC_02571*), 9 genes were involved in pathogenicity (*MAC_07957*, *MAC_05899*, *MAC_06622*, *MAC_06606*, *MAC_06944*, *MAC_08413*, *MAC_05384*, *MAC_08685* and *MAC_05385*), 6 genes were involved in growth and development (*MAC_07957*, *MAC_06622*, *MAC_02934*, *MAC_06944*, *MAC_09145* and *MAC_08685*) and 3 genes were involved in stress tolerance (*MAC_07958*, *MAC_04217* and *MAC_05385*) (Appendix A). Further analysis revealed that *MAC_09768* (Figure 5D), a DEG involved in the RNA-seq data, encoding an amino acid permease, which was homologous to *Saccharomyces cerevisiae* amino acid permease GAP1 (NP_012965.3). GAP1 is an important marker protein to respond to NCR and a main regulator of yeast plasma membrane in nitrogen starvation condition [35,36], and could interact with AreA in *A. nidulans* [37]. Therefore, we reasoned that the involvement of *MaNCP1* in regulating conidiation may be related to the NCR pathway.

## 4. Discussion

Zinc finger proteins are an important class of multifunctional regulators, in which the C2H2 zinc finger proteins have been reported in many fungi and play crucial roles in the regulation of conidiation, such as amdX [38] and nsdC [39] in *A. nidulans*, StuA in *Acremonium chrysogenum* [40], PacC [41] and Msn2 [42] in *M. acridum*. Previous studies have shown that MaNCP1 regulates the conidiation pattern [30] and virulence in *M. acridum* [31]. *MaNCP1* homologous gene *MGG_07339* regulates the conidiation and virulence in *Magnaporthe oryzae* [43]. In this study, we revealed to the role of MaNCP1 in the regulation of conidiation by affecting nitrate assimilation in *M. acridum*.

Previous study showed that *MaNCP1* regulates the nitrogen metabolism, and the DEGs of WT vs. ∆*MaNCP1* (grown on SYA media) showed that 10 genes (*MAC_00595*, *MAC_02196*, *MAC_02717*, *MAC_04470*, *MAC_05894*, *MAC_06018*, *MAC_07958*, *MAC_07959*, *MAC_09703* and *MAC_00141*) among the 45 annotated DEGs are involved in amino acid metabolism, which encoding cytochrome P450, NmrA family transcriptional regulator, catalase, oxidoreductase, major facilitator superfamily protein, tyrosinase, hydantoinase/oxoprolinase, FAD binding domain-containing protein and hypothetical protein, respectively [30]. Here, we had shown that the change in conidial yield caused by *MaNCP1* was reversible by increasing the nitrate content in SYA medium, suggesting that the conidiation mediated by *MaNCP1* was related to nitrate metabolism. Moreover, the transcription levels of genes related to the nitrate assimilation pathway were all up- or down-regulated in Δ*MaNCP1* background. Furthermore, our results also showed that *MaNCP1* affected the utilization of nitrate, ammonium, Gln and Glu, and the utilization of Gln may cause the change in secondary metabolites, resulting in the color change in the solid medium or liquid fermentation broth that inoculated with ∆*MaNCP1* strain. RNA-seq analysis under SYA+N condition showed that MaNCP1 involved in multiple metabolic pathways, the amino acid permease *MAC_09768*, a DEG, is homologous to the amino acid permease GAP1 in *S. cerevisiae*, which is regulated by the Gln3 and Gat1 in *S. cerevisiae* [44]. It has been reported that the GATA transcription factor AreA, which function is similar to Gln3/Gat1, interacts with Gap1 in *A. nidulans* [37]. In this study, we confirmed that *MaAreA* was a direct target gene of MaNCP1. Many studies have confirmed that *AreA* is a core regulator in nitrogen metabolism [45,46,47,48,49].

Furthermore, it is confirmed that *MaNmrA* is a further downstream target gene of MaNCP1 and its expression level is significantly down-regulated in Δ*MaNCP1* background [30], which is similar to *MaAreA*. In *A. nidulans*, NmrA can interact with AreA to form NmrA-AreA heterodimer and inhibit the utilization of alternative nitrogen sources (i.e., nitrate and nitrite) under sufficiently nitrogen source conditions by modulating the activity of nitrate and/or nitrite reductase. If not, it will not form NmrA-AreA complex and promoting the utilization of alternative nitrogen sources [28,50]. Although study has shown that SYA medium does not provide a classical NCR condition [30]. In general, the extremely significant down-regulation of *MaNmrA* will alleviates the inhibition on *MaAreA* in Δ*MaNCP1* background. In addition, the additional nitrate did not induce the expression of *MaAreA*, and the expression of the core genes (*MaNR* and *MaNiR*) in nitrate assimilation were significantly reduced in the absence of *MaNCP1* whether under SYA or SYA+N conditions. Under SYA condition, deletion of *MaNCP1* significantly reduced the conidial yield, however, which could be restored when exogenous nitrate was added into SYA medium. Studies have shown that nitric oxide (NO), a by-product in nitrate metabolism, is involved in regulating conidiation in *Neurospora crassa* [51] and *Coniothyrium minitans* [52]. However, excessive NO accumulation can cause the cellular nitrooxidative stress, and our previous study shows that deletion of *MaNCP1* can reduce the intracellular NO content and enhance the expression of the flavohemoglobin genes, *MaFhb1* and *MaFhb2* [30], which play indispensable roles in catalyzing the redox process of NO [53,54]. Taken together, our results showed that MaNCP1 may be involved in the NCR pathway by acting on *MaAreA* and *MaNmrA* to regulate nitrate metabolism and conidiation, suggesting that MaNCP1 play important homeostatic roles in fungal nitrate metabolism and nitrogen utilization.

## 5. Conclusions

In summary, the results of MaNCP1 regulating the conidiation in *M. acridum* will provide candidate genes and theoretical guidance for improving the conidial quality and yield to reduce the production cost and maintain the stable efficiency of mycoinsecticides.

## Figures and Tables

**Figure 1 jof-08-00942-f001:**
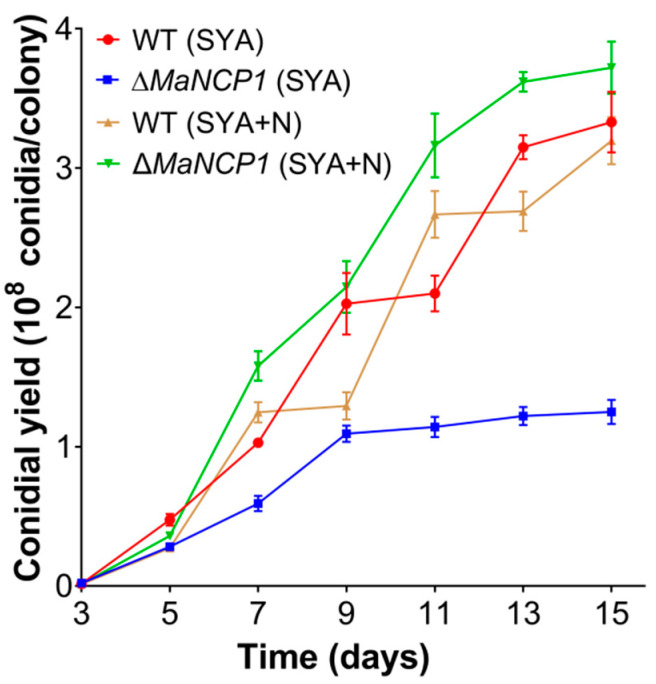
Conidial yields of the WT and ∆*MaNCP1* strains under SYA or SYA+N conditions. SYA+N, SYA supplemented with sodium nitrate.

**Figure 2 jof-08-00942-f002:**
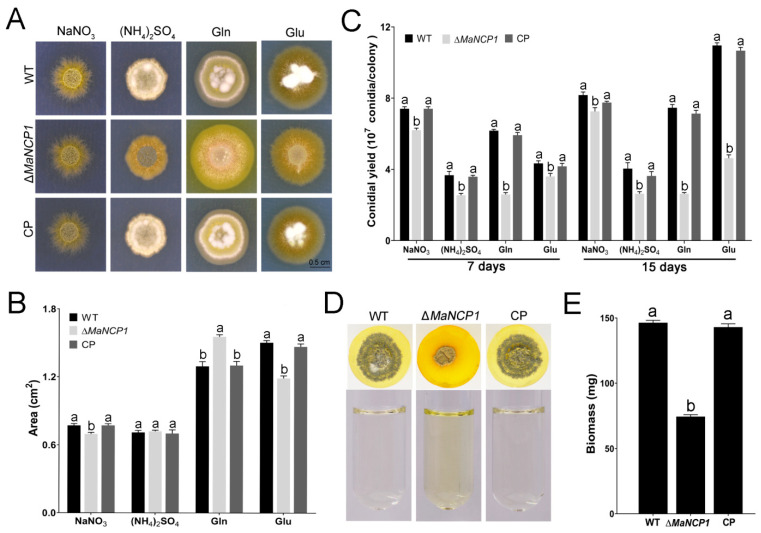
*MaNCP1* affects the nitrate assimilation. Colony morphology (**A**) and colony areas (**B**) of the fungal strains grown on the modified CZA medium at 28 °C for 7 days. The concentration of all nitrogen sources was 10 mM. (**C**) Conidial yield of each strain grown on the modified CZA medium and cultured for 7 or 15 d, respectively. (**D**) Colony morphology of each strain cultured in the 24-well plates for 7 d (**Top**), and fermentation liquor of the fungal strains inoculated in CZB (CZA without agar) with 220 rpm for 48 h (**Bottom**). Gln (10 mM) was acted as the sole nitrogen source in CZB liquid medium. (**E**) Biomass of the liquid fermentation for 48 h. Gln, Glutamine. Glu, Glutamate. Lowercase letters indicate significant difference at *p* < 0.05 (Tukey’s HSD at the same treatment and the letters indicate a comparison between strains cultivated on the same modified CZA.

**Figure 3 jof-08-00942-f003:**
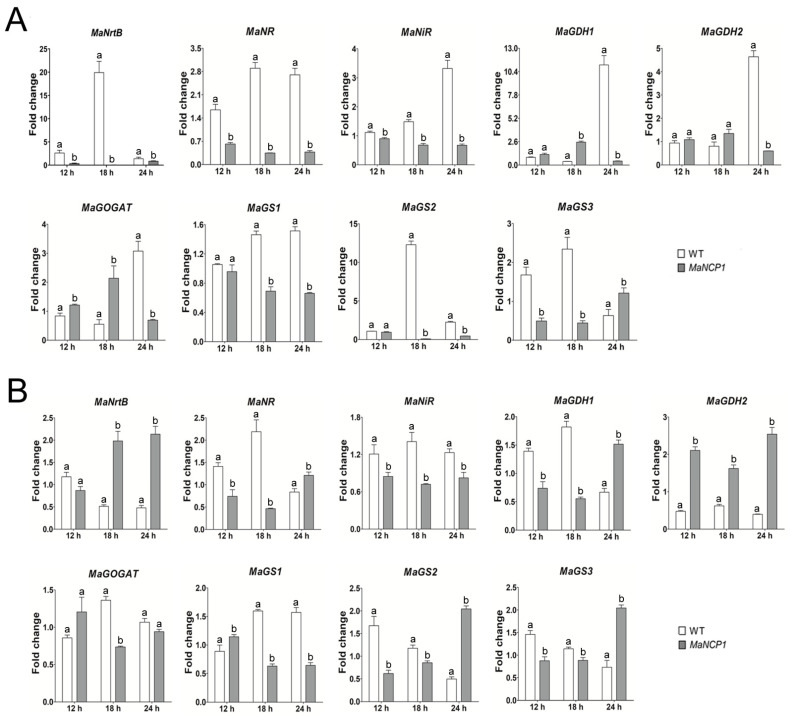
*MaNCP1* regulated the expression of nitrate and ammonium assimilation genes. Transcription level analysis of genes in nitrate and ammonium assimilation pathway under SYA (**A**) and SYA + N (**B**) conditions. Fungal strains were cultured on SYA and/or SYA+N media for 12, 18 or 24 h. Lowercase letters indicate significant difference at *p* < 0.05 (T-text).

**Figure 4 jof-08-00942-f004:**
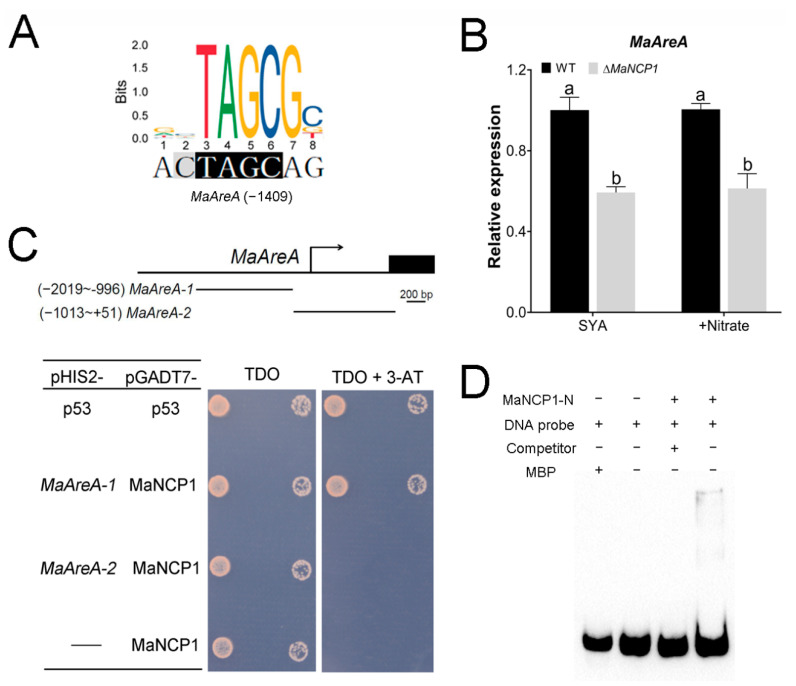
*MaAreA* is a target gene of MaNCP1. (**A**) Putative binding site prediction of MaNCP1. The potential *cis-element* was identified via JASPAR 2020 database [33]. (**B**) Transcription level analysis of *MaAreA*. Fungal strains were spread on SYA or SYA+N media and cultured at 28 °C for 18 h. Lowercase letters indicate significant difference at *p* < 0.05 (T-text). (**C**) Yeast one-hybrid assay. TDO, SD/-Leu/-Trp/-His. The concentration of 3-AT was 8 mM. (**D**) The EMSA analysis. MBP, MBP-tag protein. MaNCP1-N, the N-terminal of MaNCP1 protein. Competitor, the unlabeled probe, which was added in a 100-fold excess. +, probe or protein added, −, probe or protein not added.

**Figure 5 jof-08-00942-f005:**
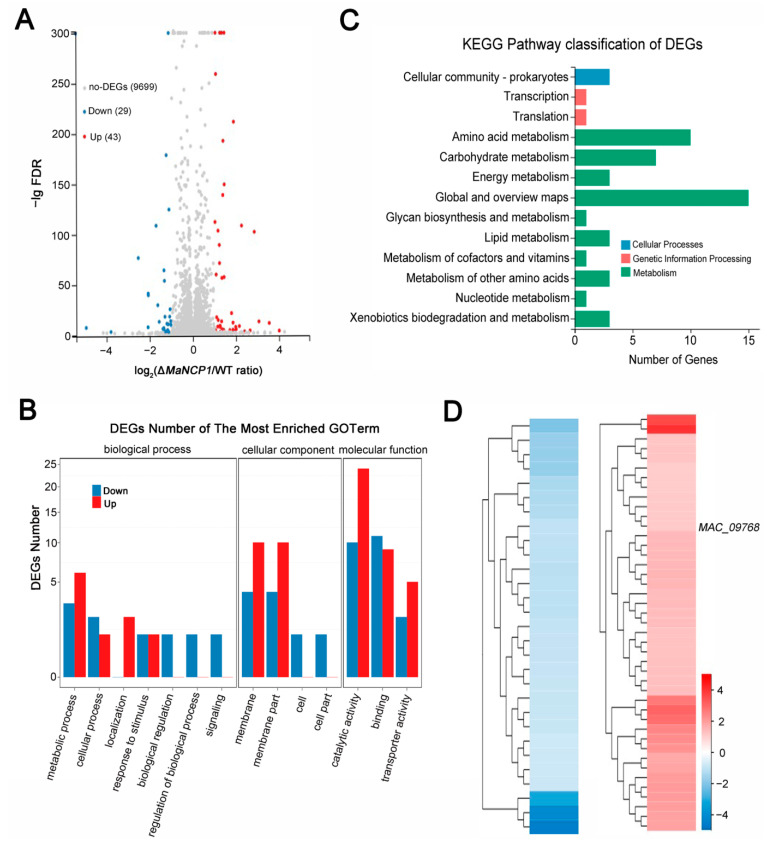
RNA-seq analysis of Δ*MaNCP1_*N-vs-WT_N. (**A**) Distributions of DEGs. (**B**) Analysis of the GO function classes. (**C**) KEGG pathway classification of DEGs. (**D**) Clustering analysis of the known DEGs.

## Data Availability

RNA-seq data had been are deposited in the NCBI BioProject database (accession No. PRJNA799900).

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
