# Peer review of "The C2H2 Zinc Finger Protein MaNCP1 Contributes to Conidiation through Governing the Nitrate Assimilation Pathway in the Entomopathogenic Fungus Metarhizium acridum"

_jof, 2022, doi:10.3390/jof8090942_

Round 1
Reviewer 1 Report
Introduction: Metarhizium acridum is an acridid-specific pathogen. This should be mentioned in the introduction. The authors only mentioned M. anisopliae, which lifestyle and host range could be much more variable than M. acridum.
Please provide the PCR efficiency for each prime used in the qPCR analysis. How many serial dilutions did the authors make to build the standard curve? The authors can use table S1 to include the PCR efficiency.
Figures 2B and 2C: The letters indicate a comparison between strains cultivated on the same modified CZA, right? Please specify that so there is no mistake with a possible comparison of the same strain among the different culture media.
Figure 3: Please indicate the statistical difference between the strains (WT and MaNCP1) in each graph for each time. Only after that, you can really say they were up or downregulated in the text. Please include in the caption the statistical test that was used.
Lines 118 and 197: for how many hours? In the text, it is just “for hours.”
Reviewer 2 Report
The manuscript "The C2H2 zinc finger protein MaNCP1 contributes to conidiation through governing the nitrate assimilation pathway in the entomopathogenic fungus Metarhizium acridum" is well written, and complements the topics previosly published for the research group.
Specific comments:
- Define the acronyms
- l.173 - Capitalized letters indicate significant difference at p < 0.05 (Tukey’s HSD) within each treatment?
Reviewer 3 Report
The Manuscript [jof-1910419] entitled (The C2H2 zinc finger protein MaNCP1 contributes to conidiation through governing the nitrate assimilation pathway in the entomopathogenic fungus Metarhizium acridum) focused on the roles of a C2H2 zinc finger protein MaNCP1 in nitrogen utilization and conidiation of Metarhizium acridum. The results stated that MaNCP1 contributes to the nitrate assimilation and conidiation of this fungus. The manuscript has good results those are introduced and written very well. Here, some comments for the authors that are considered as minor revision
1- Lines 23: [It could be concluded that] instead of [In summary, our data revealed that]
2- Line 76 to 79: delete [Our results indicated ……. Pathway.]. these are results. And explain the objective in more details.
3- Line 96: [Two microliters of conidial]
4- Line 123: [(MAC_09584) and the glyceraldehyde …]
5- Lines 137-140: delete [Previous study shows …… nitrate]. They are discussion and methods.
6- Lines 151-153: Delete [To analyze ……………, respectively]
7- Figure 2: indicate that [Capitalized letters indicate significant difference at p < 0.05 (Tukey’s HSD at the same treatment).
8- Data of Figure 4B: they analysed with one-way ANOVA or T-test? They only 2 treatments inside SYA or +nitrate.
9- Line 266: [we revealed to the role]
10- Line 268: [showed] instead of [shows]
11- Lines 314-317: add title of [conclusion]
